# Titica Vine Fiber (*Heteropsis flexuosa*): A Hidden Amazon Fiber with Potential Applications as Reinforcement in Polymer Matrix Composites

Juliana dos Santos Carneiro da Cunha [1,*], Lucio Fabio Cassiano Nascimento [1], Fernanda Santos da Luz [1], Fabio da Costa Garcia Filho [1,2], Michelle Souza Oliveira [1] and Sergio Neves Monteiro [1]

1 Department of Materials Science, Military Institute of Engineering—IME, Praça General Tibúrcio, 80, Urca, Rio de Janeiro 22290-270, Brazil

2 Department of Mechanical and Aerospace Engineering, University of California San Diego—UCSD, La Jolla, CA 92093, USA

* Correspondence: julianasccunha@gmail.com

**Abstract:** The titica vine fiber (TVF) (*Heteropsis flexuosa*) is a natural lignocellulose fiber (NLF) from the Amazon rainforest that was, for the first time, investigated in terms of its basic properties such as dimensions, porosity, and density as well as its chemical composition, moisture content, crystallinity, and microfibrillar angle. In this study, the apparent density of TVF was determined as one of the lowest-ever reported for NLFs). Using both the geometric method and Archimedes' principle, density values in the range of 0.5–0.6 g/cm$^3$ were obtained. The moisture content was measured as around 11%, which is in accordance with the commonly reported values for NLFs. The TVF exhibited a high porosity, approximately 70%, which was confirmed by SEM images, where a highly porous morphological structure associated with the presence of many voids and lumens was observed. The crystallinity index and microfibrillar angle were determined as 78% and 7.95°, respectively, which are of interest for a stiff NLF. A preliminary assessment on the mechanical properties of the TVFs revealed a tensile strength, Young's modulus, and elongation of 26 MPa, 1 GPa, and 7.4%, respectively. Furthermore, the fiber presented a critical length of 7.62 mm in epoxy matrix and an interfacial shear strength of 0.97 MPa. These results suggest the TVFs might favors applications where lighter materials with intermediate properties are required.

**Keywords:** titica vine fibers; natural lignocellulosic fibers; density; porosity; X-ray diffraction; mechanical properties; morphological characterization



## 1. Introduction

The industrial sector, in past decades, has gone through a transition process that affected its main production systems [1]. Growing interest in the development and manufacture of green composites has occurred due to new environmental regulations and high consumer demand to minimize the use of synthetic fibers in petroleum-based polymer composites [2–4]. In this respect, natural lignocellulosic fibers (NLFs) are promising alternatives as filler for polymer composites. Indeed, they are able to improve the efficiency of many mechanisms through not only their individual characteristics but also their interaction with polymer matrices [5]. In addition, they are low-density, biodegradable, recyclable, and cost-effective materials that possess relatively high specific properties [2–4,6].

NLFs and their fabrics are applied industrially in various sectors, for production of ropes, handicrafts, textiles, and furniture as well as automobile, sports, and civil construction parts reinforcement for composite materials [7–9]. In recent years, different NLFs have been investigated for these applications, such as bamboo [10], malva [11], buriti [12], and curaua [13] among others. In particular, numerous studies have been carried out with the

aim of considering NLFs as an integral part of the middle layer in a multilayer armor system (MAS), aiming at personal protection against 7.62 mm and 0.22 in ammunition [14–17].

The structure of NLFs might be associated with that of a natural composite material, which is fundamentally formed by two distinct phases: one amorphous and one crystalline [9,18]. The cell wall consists mainly of cellulose microfibrils, which can be seen as the reinforcing phase, surrounded by a matrix of lignin, which is a totally amorphous polymer [18,19]. Cellulose microfibrils are arranged along the entire length of the fiber, allowing maximum tensile and flexural strength, in addition to promoting stiffness [18]. NLFs' mechanical properties also depend on the angle formed between the fiber axis and the orientation of the microfibrils, called the microfibrillar angle (MFA) [20]. There is a strong correlation between the MFA and the Young's modulus of a fiber [20,21].

It is known that chemical, physical, and morphological properties influence the mechanical properties of NLFs [22]. In their research, Thakur et al. [19] discussed how the type and degree of cellulose polymerization, as well as the amount of the MFA, affected the mechanical properties of NLFs. Reis et al. [23] verified the relationship between the low MFA of guaruman fibers and the high tensile strength. Wang et al. [24], characterizing kenaf fibers, reported that the chemical composition, crystallinity, and degree of orientation have a strong influence on their tensile strength. Monteiro et al. [25] investigated the correlation of tensile strength of curaua, sisal, and ramie fibers of different diameters and found that thin fibers presented comparatively higher values. According to Luz et al. [26], among the microstructural defects, porosity can play a significant role in the properties of NLFs. When studying coconut fibers, they observed that porosity increases directly with diameter; consequently, fibers with smaller diameters are suitable to guarantee superior properties in reinforced composites.

Located in a tropical region, with vast forests and possessing favorable climate for plants, Brazil is a country with great variety of NLFs [27]. Titica vine (*Heteropsis flexuosa*), belonging to the *Araceae* family, is a typical plant from the Amazon region that occurs in non-flooded areas [28]. It is a family with a little-studied taxonomically and rarely, in phytosociological and floristic surveys, are its representative species included [29]. Popularly called a vine, its fibers are extracted from aerial roots that are thrown towards the ground, and, when they reach the surface, they are thick, woody, resistant, and durable [30], as illustrated in Figure 1. Found in Brazil, as well as Venezuela, Guyana, and Suriname [28], there is a large market for products made with TVFs [31].

In South America, the people of the forest use TVF to make furniture, baskets, brooms, and handicrafts in general [31]. However, TVF is little-known as a reinforcement in polymeric composites for application in engineering. Cunha et al. [5] carried out a preliminary study incorporating TVF in epoxy matrix and observed that the increase in fiber volume percentage provided a significant improvement in the thermal properties and in the Charpy and Izod impact resistance of the material. Despite this pioneering research, a broad study on the individual characteristics of the fiber has not yet been carried out.

In view of the little-explored potential of TVF, the present work aims to investigate it in terms of diameter, density, porosity, moisture content, chemical interactions through Fourier transform infrared spectroscopy (FTIR), crystallinity index, and MFA through X-ray diffraction analysis, as well as mechanical properties by means of tensile tests and, finally, to evaluate the morphological surface by scanning electron microscopy.

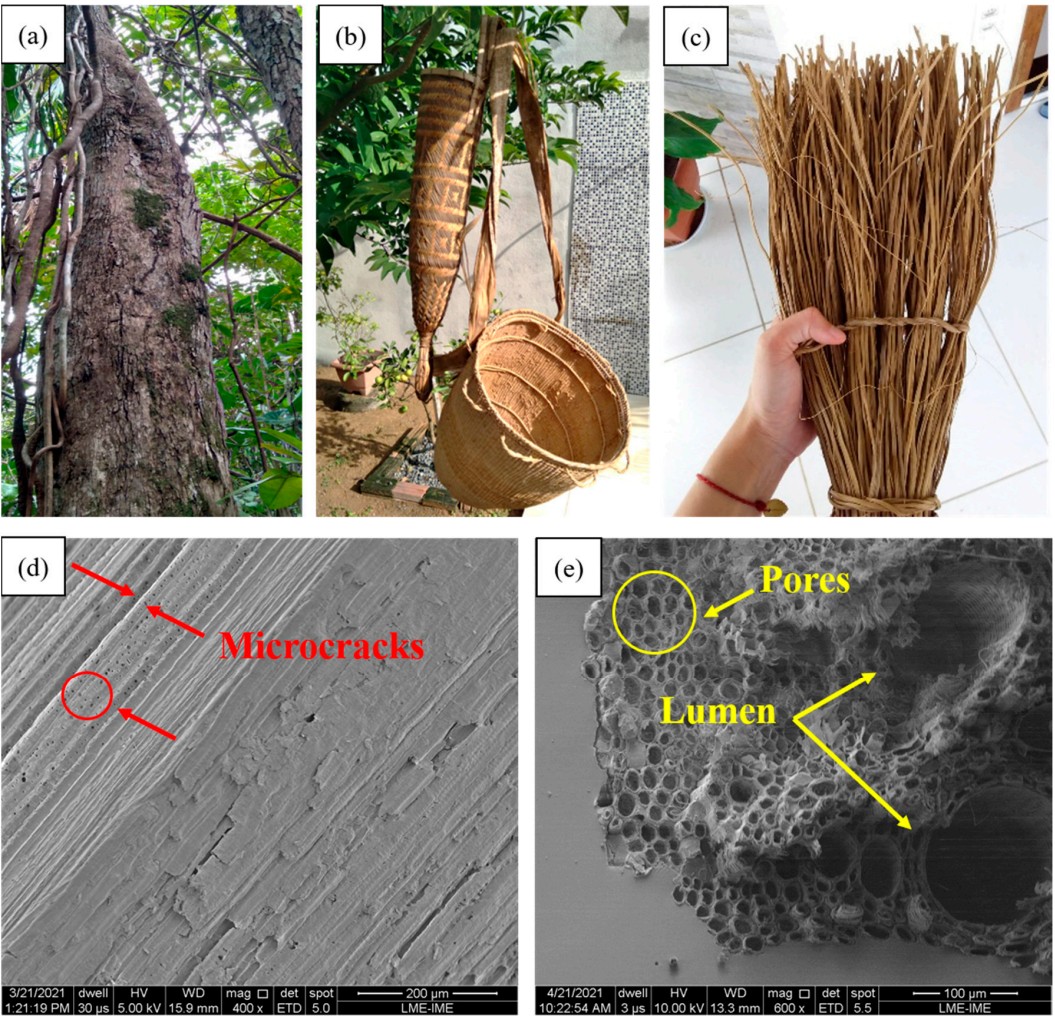

**Figure 1.** (**a**) Titica vine roots as found in the forest; (**b**,**c**) artifacts manufactured with TVFs; (**d**) SEM image of the longitudinal section of the TVF indicating defects; (**e**) SEM image of the TVF cross section indicating regions of lumens and pores.

## 2. Materials and Methods

### 2.1. TVFs Processing

Titica vine roots were obtained from a local market in the city of Boa Vista, in the state of Roraima, in the Amazon region of Brazil. Splints were cut to an approximate length of 15 cm, which the TVFs bundle were extracted from. Figure 2 presents the TVFs. Afterwards, the TVFs were cleaned in running water and dried in an air oven at 60 °C for 24 h.

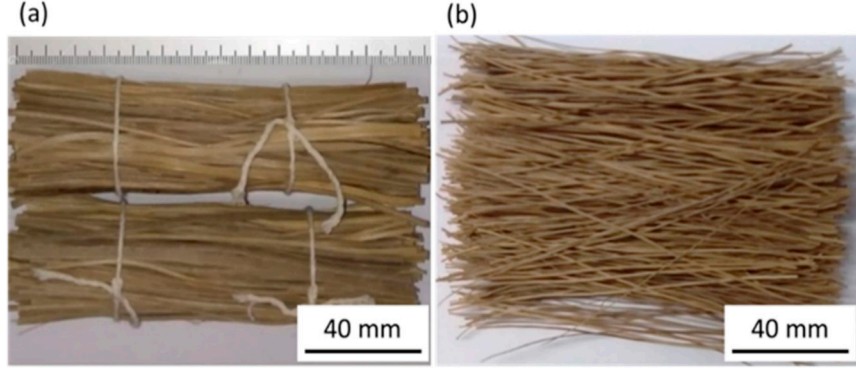

**Figure 2.** Titica vine as (**a**) splints and (**b**) bundle.

### 2.2. Physical Properties of the TVFs

2.2.1. Density Determination

Three different methods were used to assess the fiber density. The first was based on the geometric calculation, where the fiber is taken as a cylinder. One hundred fibers were weighted, using an electronic balance, model GEHAKA AG-200, with 0.0001 g precision, and had their length and diameter determined with a precision caliper of 0.01 mm and optical microscopy observations, in a model OLYMPUS BX53M, with $5\times$ magnification.

The second method was based on Archimedes' principle. Tests were carried out in accordance with ASTM D3800 standard [32], where six samples containing bundles were immersed in gasoline. The density of the material can be calculated as per:

$$\rho_f = \frac{(M_3 - M_1)\rho_l}{(M_3 - M_1) - (M_4 - M_2)} \tag{1}$$

where $\rho_f$ is the density fiber; $\rho_l$ is the density of the liquid, in this case gasoline, 0.72–0.78 g/cm$^3$ (the value of 0.77 g/cm$^3$ was taken in this investigation); $M_1$ weight of suspension wire in air; $M_2$ the weight of suspension wire in liquid; $M_3$ the weight of suspension wire plus item with density that was to be determined in air; and $M_4$ weight of suspension wire plus item with density that was to be determined in liquid.

The third method used to determine the TVFs density was through a helium gas pycnometer. The analysis was performed by a Micromeritic AccuPyc 1330, in accordance with ASTM D4892 standard [33].

2.2.2. Porosity Determination

Total ($P_T$), open ($P_O$), and closed ($P_C$) porosity of the TVFs were calculated using the geometric density ($\rho_g$), apparent density ($\rho_{ap}$), and absolute density ($\rho_{abs}$), as previously described by Luz et al. [26]. The calculations followed Equations (2)–(4):

$$P_T = \left(1 - \frac{\rho_g}{\rho_{abs}}\right) \tag{2}$$

$$P_O = \left(1 - \frac{\rho_g}{\rho_{ap}}\right) \tag{3}$$

$$P_C = P_T - P_O \tag{4}$$

The absolute density was considered to be obtained by gas pycnometry and the apparent density obtained by Archimedes' principle [34].

2.2.3. Moisture Content

The moisture content of the fibers was determined according to ASTM D1348 standard [35]. This method is based on the loss of moisture from an oven at a temperature of 105 °C. Six samples of TVFs were turn into 1 g of powder and taken to the oven for two hours. After this time, the samples were weighed again, and the steps were repeated every half hour until the mass loss is stabilized. The moisture content is determined as follows:

$$\%MC = \frac{IFM - DFM}{DFM} \times 100 \tag{5}$$

where IFM is the initial fiber mass, and DFM is the dry fiber mass.

### 2.3. Fiber Composition and Structure

2.3.1. Fourier Transform Infrared Spectroscopy (FTIR)

FTIR analysis was performed on Shimadzu IRPrestige-21 equipment (Tokyo, Japan). The fibers were ground in the powder condition, so that a pellet with KBr could be produced.

The scan was performed in a spectral range from 4000 to 400 cm$^{-1}$. Transmittance spectra were generated as a function of wave number.

### 2.3.2. X-ray Diffraction (XRD)

XRD analysis was performed on PANalytics equipment, model X'Pert PRO, with Cobalt radiation (1.789 Å), power 40 mA $\times$ 40 kV, and scanning range from 10 to 75°. To calculate the crystallinity index of TVFs, the methodology presented by Segal et al. [36] was used, in which the maximum intensity obtained in the diffractogram was used for the area of the amorphous (101) and crystalline (002) peaks. These two peaks are related to the amorphous ($I_{am}$) and crystalline ($I_{crys}$) phases, respectively. The fiber crystallinity index was calculated from Equation (6).

The microfibrillar angle (MFA) was calculated using the methodology proposed by Donaldson [37] and Sarén and Serimaa [38] through the deconvolution of the crystalline cellulose peak (002). Based on the diffractogram, the MFA is estimated from three curves resulting from the peak (002), i.e., the Gaussian and its first and second derivatives. Through this methodology, it is possible to obtain the parameter "T", according to Equation (7), which is half the distance between the intersections of the tangents at the inflection points of the curves.

$$Crl = \frac{I_{crys} - I_{am}}{I_{crys}} \times 100 \tag{6}$$

$$MFA = -12.19 \times T^3 + 113.67 * T^2 - 348.40 \times T + 358.09 \tag{7}$$

### 2.4. Tensile Tests

Tensile tests of the TVFs were performed on five samples for each diameter range, according to ASTM D3822 standard [39], in an Instron 3365 universal machine, with a crosshead speed of 0.4 mm/min. Tensile strength, Young's modulus, and deformation at rupture were calculated from the stress–strain curve. In addition, the tensile strength results were statistically evaluated through Weibull parameters as well as by analysis of variance (ANOVA). The latter, in order to verify if there was an influence of the diameter on the property.

### 2.5. Pullout Tests

The pullout test estimates the shear strength interface, based on a composite sample with a single fiber. The method involves pulling a single fiber partially embedded in a block of polymer matrix [40]. In this case, the matrix used was of the epoxy type, bisphenol A diglycidyl ether (DGEBA) with 13% triethylenetetramine (TETA) hardener, both supplied by the Epoxyfiber (Rio de Janeiro, Brazil). From the pullout stress versus embedded length graph, the interfacial shear strength ($\tau$) and critical length ($L_c$) were obtained [41], according to Equation (8).

$$\tau = \frac{d\sigma_f}{2L_c} \tag{8}$$

where d is the average fiber diameter; $\sigma_f$ the TVF tensile strength. The tests were performed on a previously mentioned Instron machine at a speed of 1 mm/min.

### 2.6. Scanning Electron Microscopy (SEM)

Morphological aspect of TVFs was investigated by scanning electron microscopy (SEM), on a Quanta FEG 250 Fei microscope model (Hillsboro, OR, USA) operating with secondary electrons at 2 and 5 kV.

## 3. Results

### 3.1. Physical Properties of the TVFs

3.1.1. Density Determination

The diameter distribution range of the TVFs is presented in Figure 3. One may notice that the fiber distribution might be regarded as a normal distribution, with an estimated mean diameter of 650.1 ± 175.7 µm. The broad range of fiber diameter, from 356.6 up to 1110.6 µm, could result in a heterogeneity of properties, as will be further discussed. Table 1 presents the average density calculated for each diameter interval. The standard deviation and the Weibull parameters, where β is the Weibull module, also called the shape parameter, and θ is the scale parameter, in this case related to the evaluated property.

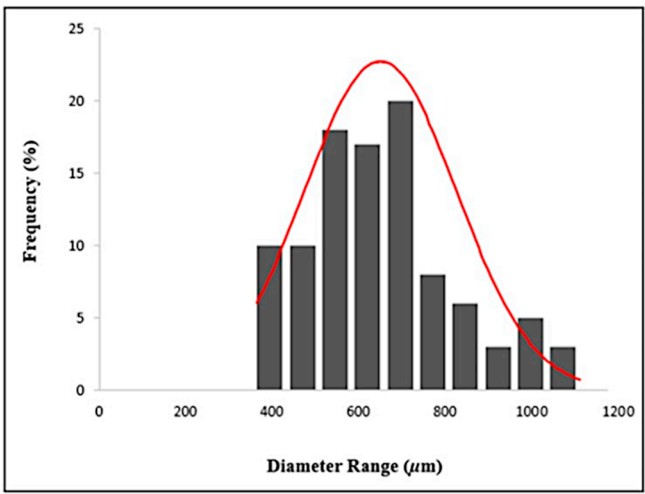

**Figure 3.** Frequency of the number of TVFs versus diameter range and their normal distribution curve.

**Table 1.** Average geometric density calculated for the different diameter intervals.

| Diameter Range (µm) | Density (g/cm$^3$) | Standard Deviation | β | θ | R$^2$ |
|---|---|---|---|---|---|
| 356.63–432.03 | 0.590 | 0.136 | 4.695 | 0.645 | 0.854 |
| 432.03–507.43 | 0.531 | 0.081 | 7.176 | 0.566 | 0.979 |
| 507.43–582.83 | 0.510 | 0.088 | 6.442 | 0.547 | 0.862 |
| 582.83–658.23 | 0.498 | 0.076 | 7.238 | 0.531 | 0.922 |
| 658.23–733.63 | 0.483 | 0.084 | 6.093 | 0.520 | 0.962 |
| 733.63–809.03 | 0.541 | 0.065 | 8.970 | 0.569 | 0.975 |
| 809.03–884.43 | 0.494 | 0.046 | 10.982 | 0.515 | 0.918 |
| 884.43–959.83 | 0.490 | 0.018 | 26.764 | 0.498 | 0.997 |
| 959.83–1035.23 | 0.445 | 0.067 | 7.413 | 0.472 | 0.950 |
| 1035.23–1110.63 | 0.439 | 0.055 | 10.563 | 0.457 | 0.938 |

It is possible to verify a certain trend of decreasing density with increasing fiber diameter. Inverse relationships such as this were also observed for tensile strength and the diameter of the other species of NLFs [42,43]. This behavior could be associated with a higher probability of internal defects, such as voids, in their microstructure, as the diameter of the fiber is increased. The mean density of the TVFs was calculated as 0.50 ± 0.07 g/cm$^3$ by the geometric method. The Weibull parameters suggested the same tendency of the property evaluated for the TVFs, in which the lowest densities are located in the largest cross-sections. It is worth noticing that the value of R$^2$ remained between 0.85–0.99, presenting a statistically acceptable data reliability.

The density of TVFs was also estimated using Archimedes' principle. Table 2 shows the values of density obtained by the TVFs according to Archimedes' principle. In this method, it was not possible to accurately measure the density differences between thinner

and thicker TVFs. According to Marchi et al. [44], this could be associated with lack of penetration of immersion fluidly gasoline into the open porosity of the fiber. Nonetheless, the average density calculated from Archimedes' principle was $0.57 \pm 0.03$ g/cm$^3$.

**Table 2.** Densities obtained by Archimedes' principle.

| Sample | Dry Sample Weight (g) | Weight of the Immersed Sample (g) | Wet Sample Weight (g) | Density (g/cm$^3$) |
|---|---|---|---|---|
| 1 | 1.801 | 0.311 | 2.658 | 0.588 |
| 2 | 2.202 | 0.400 | 3.258 | 0.590 |
| 3 | 2.178 | 0.276 | 3.231 | 0.565 |
| 4 | 0.716 | 0.134 | 1.158 | 0.536 |
| 5 | 0.828 | 0.169 | 1.222 | 0.603 |
| 6 | 0.679 | 0.132 | 1.120 | 0.527 |

One should notice that, by both methodologies, the TVF density is found to be among the lowest ever reported for NLFs that are commonly used as reinforcement of polymer matrix composites, such as bamboo (1.03–1.21 g/cm$^3$), sisal (1.26–1.50 g/cm$^3$), ramie (1.50 g/cm$^3$), jute (1.30–1.45 g/cm$^3$), curaua (0.57–0.92 g/cm$^3$), and piassava (1.10–1.45 g/cm$^3$) [9]. There are reports in the literature of low-density NLFs that have excellent mechanical properties. This is the case of guaruman (0.37–0.66 g/cm$^3$) [23], curaua (0.57–0.92 g/cm$^3$) [9], and bagasse (0.34–0.49 g/cm$^3$) [9], which have tensile strengths of 614 MPa, 117 to 3000 MPa, and 135 to 222 MPa, respectively. Furthermore, in these studies, the Young's modulus of these fibers ranged from 15 to 80 MPa.

The gas pycnometry test disclosed an absolute density value for the TVFs of $1.62 \pm 0.07$ g/cm$^3$. The higher value determined by this technique could be related to the high sensitivity of the technique as well as to the climatological conditions during the test, which may have favored the absorption of water into the TVFs. In addition, part of the difference between absolute (gas pycnometry) and apparent (geometric method and Archimedes' principle) density could be associated with the presence of open porosity in NLFs. The gas used in the pycnometer analysis, helium, is only capable of filling open-pore cavities, thus closed pores might be disregarded during this analysis [45]. Higher absolute density values by pycnometry compared to other techniques were also observed by Oliveira et al. [46] and Fiore et al. [47] for the evaluation of tucum and artichoke fibers, respectively.

The relatively high absolute density of TVF obtained by this technique can also be justified by the density presented by neat cellulose, which is around 1.6 g/cm$^3$ [34,48]. Moreover, most of the NLFs used, as promising reinforcement for polymer matrix composites for engineering application, exhibit densities in the range of 1.4–1.6 g/cm$^3$ [49].

### 3.1.2. Porosity Determination

The average of total, open, and closed porosity for each diameter interval were estimated, as shown in Table 3.

Only for the first range, the proposed calculation method was not applicable or even possible, once the apparent density was greater than that obtained geometrically. One should notice that there is a tendency. The larger the diameter is, the greater the open and total porosity of the fiber. This behavior, once again, might be associated with the greater number of defects that can be found in thicker fibers [50,51]. The average of total, open, and closed porosities for the entire set of 100 fibers was 69.01%, 11.62%, and 57.39%, respectively. It is possible to note that TVFs presented a higher amount of closed porosity than open porosity, i.e., there is a greater amount of air cavities and voids on the inner part of the fiber, with a large presence of lumens.

**Table 3.** Percentage of total, open, and closed porosity for the different diameter intervals.

| Diameter Range (μm) | Total Porosity (%) | Open Porosity (%) | Close Porosity (%) |
|---|---|---|---|
| 356.63–432.03 | - | - | - |
| 432.03–507.43 | 67.222 | 6.514 | 60.708 |
| 507.43–582.83 | 68.519 | 10.211 | 58.307 |
| 582.83–658.23 | 69.259 | 12.324 | 56.935 |
| 658.23–733.63 | 70.185 | 14.965 | 55.220 |
| 733.63–809.03 | 66.605 | 4.754 | 61.851 |
| 809.03–884.43 | 69.506 | 13.028 | 56.478 |
| 884.43–959.83 | 69.753 | 13.732 | 56.021 |
| 959.83–1035.23 | 72.531 | 21.655 | 50.876 |
| 1035.23–1110.63 | 72.901 | 22.711 | 50.190 |

These results reveal that, indeed, the TVFs presents a highly porous microstructure, with total porosity that is higher than other NLFs such as jute (40.08%) [52], aloe vera (47.86%) [52], and coconut (45%) [26], while being similar to tucum (73%) [46]. The significant amount of porosity present in TVF is reflected in the lower apparent density, which makes it a promising reinforcement for composite materials, in which applications require lightness, for instance, for ballistic protection.

### 3.1.3. Moisture Content

The moisture content of NLFs is a characteristic that depends on the fraction of non-crystalline parts and the concentration of voids in the fiber [53]. In general, NLFs have a hydrophilic character and most resins are hydrophobic [22,51]. The understanding of the characteristic of the fiber is mandatory for determining the compatibility between fibers and a possible polymer matrix. The TVFs exhibit an average moisture content of $11.4 \pm 0.8$ %. This value is similar to those reported for traditional NLFs such as flax (12%) [54], ramie (15%) [54], jute (13.75%) [55], coconut (13%) [54], and sisal (11%) [56]. It is worth mentioning that the variability of the moisture content is strongly influenced by factors such as plant age, soil, storage, etc. [9]. The relatively low moisture content exhibited by the TVFs may favor their application as a reinforcement in a polymeric matrix. Indeed, the less hydrophilic the TVFs are, the lower the percentage of water inside the matrix, which strongly influences the interfacial adhesion between a fiber and the matrix [57].

### *3.2. Fiber Composition and Structure*

### 3.2.1. Fourier Transform Infrared Spectroscopy (FTIR)

FTIR analysis was performed on TVFs to characterize the fundamental chemical components found in NLFs. It is known that the basic components in NLFs are cellulose, lignin, and hemicellulose, as well as other components such as alkanes, esters, aromatics, ketones, and alcohols, where the presence of oxygen is often observed [58]. From the data obtained by the analysis, a fiber was plotted and is presented in Figure 4.

The broad band, which can be observed at 3433 cm$^{-1}$, is characteristic of all lignocellulosic materials and can be attributed to the axial deformation of the O-H group [59]. At 2924 cm$^{-1}$ and 1368 cm$^{-1}$, C-H stretching is common generally observed in cellulose and hemicellulose molecules [60,61]. At 1735 cm$^{-1}$, there is a band referring to the axial deformation of C=O, corresponding to the ester bonds of the lignin carboxylic group and of the uronic and acetyl ester groups of the hemicelluloses [62].

The band at 1651 cm$^{-1}$ is attributed to the carbonyl group of the hemicellulose ester and lignin aldehydes [63]. At 1512 cm$^{-1}$ it is possibly related to the C=C stretch bonds associated with lignin [64,65]. The signal at 1265 cm$^{-1}$ corresponds to the movement of the vibrations of the C-O bonds of the acetyl groups present in hemicellulose and lignin [65]. Another frequent band in NLFs is at 1041 cm$^{-1}$, which may be associated with the C-H groups and C-O deformations present in hemicellulose and cellulose [62]. Finally, the band

at 594 cm$^{-1}$ refers to the C-C (aromatic) deformations belonging to cellulose, hemicellulose, and lignin [66,67].

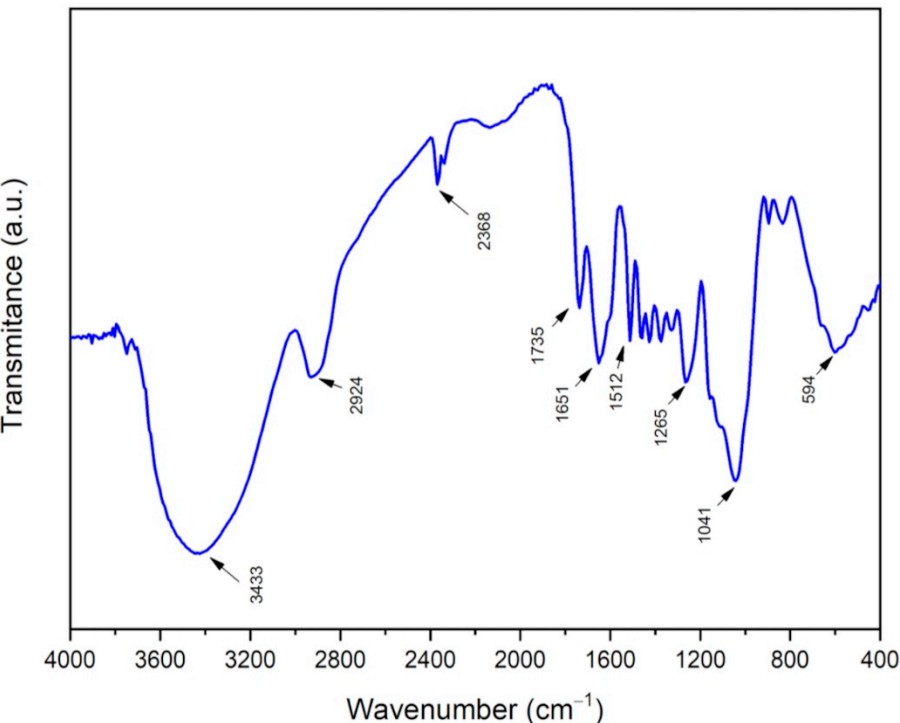

**Figure 4.** TVF FTIR spectrum.

### 3.2.2. X-ray Diffraction (XRD)

For this analysis, vertically aligned TVFs were used in the sample holder of the equipment. Figure 5 presents the diffractogram pattern obtained for the TVF.

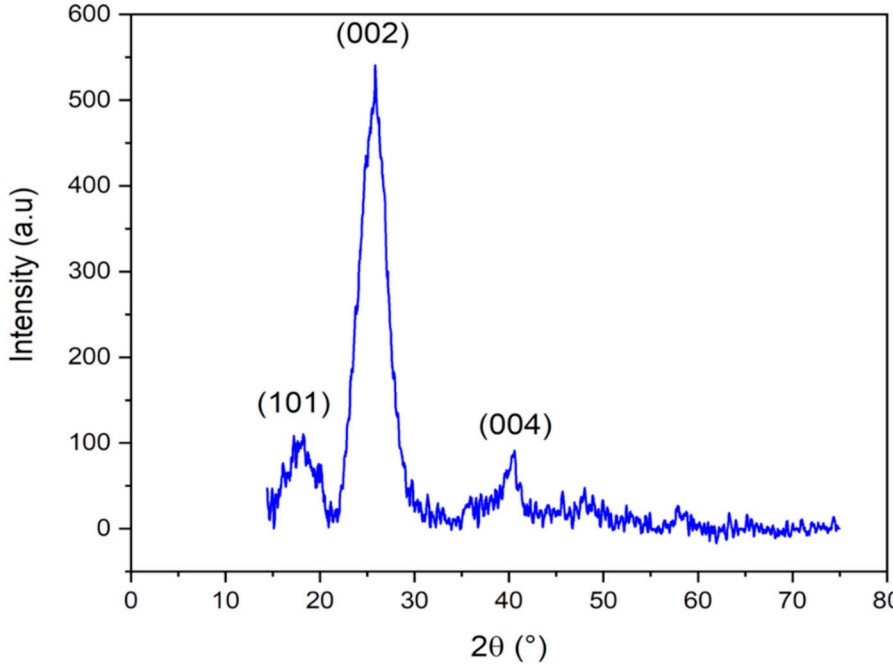

**Figure 5.** TVF XRD pattern.

It is observed that the TVFs present a semi-crystalline behavior, with emphasis on the planes (101), (002), and (004). The crystallinity index was obtained using Equation (6), which relates the interference intensity in the crystalline plane (002), corresponding to the peak at $2\theta = 25.84°$, and the scattering of the amorphous region in the (101) plane, corresponding to the $2\theta$ peak $= 18.22°$. Using the method proposed by Segal et al. [38], the crystalline index of the TVF was calculated as 78.3%. In addition, the methodology proposed by Mwaikambo and Ansell [34] was used to quantify the amount of crystalline and amorphous cellulose, which was shown to be 39%. From the diffraction pattern presented in Figure 5, it was possible to estimate the microfibrillar angle (MFA) isolating the peak, attributed to the crystalline phase in the (002) plane and based on mathematical methods. Figure 6 presents the graphical solution that relates the peak (002), adjusted by a Gaussian curve and its first- and second-order derivatives.

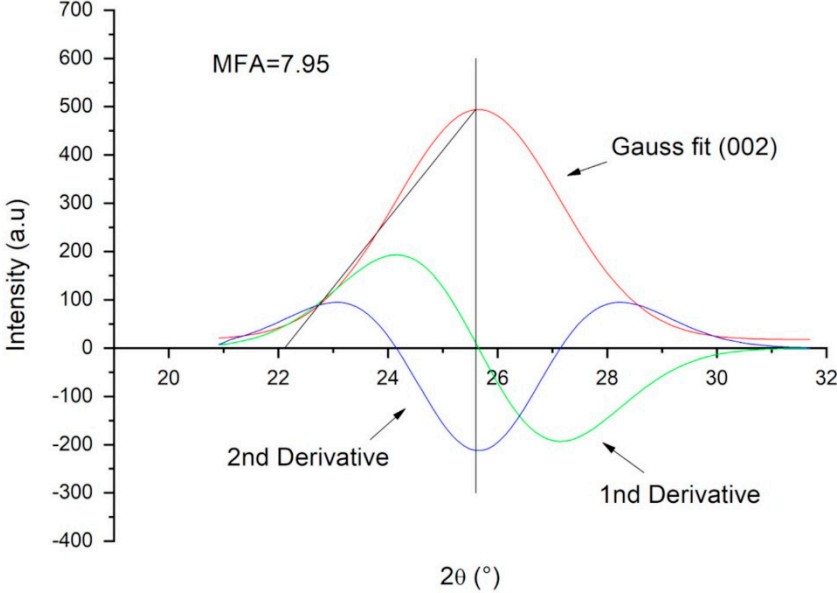

**Figure 6.** TVF microfibrillar angle.

The MFA of the TVF was calculated as 7.95°. It is reported that the MFA, as a microfibrillar orientation indicator, especially affects the fiber axial strength properties [20]. A low MFA makes the fiber highly anisotropic, and this leads to relatively low transverse mechanical properties. As a result, cell wall structure rules the physical properties of NLFs. Table 4 compares the properties obtained for the TVFs with other NLFs in terms of apparent density obtained by the geometric method or Archimedes' principle, crystallinity, cellulose, and MFA.

**Table 4.** Density, crystallinity index, cellulose content and microfibrillar angle of TVFs compared to other NLFs.

| NLFs | Density (g/cm³) | Crystallinity Index (%) | Cellulose (%) | MFA (°) | Reference |
|---|---|---|---|---|---|
| Titica Vine | 0.50 | 78.3 | 39 | 7.95 | PW |
| Buriti | 1.31 | 63.1 | 58 | 7 | [12] |
| Sisal | 1.33 | 72.2 | 73 | 20 | [68,69] |
| Jute | 1.40 | 73.4 | 72 | 8 | [67–69] |
| Coir | 1.20 | 43.5 | 36 | 51 | [68–70] |
| Flax | 1.38 | 72.2 | 75 | 10 | [68,70–73] |
| Ramie | 1.50 | 76.4 | 68 | 7.5 | [69,71,73,74] |

Analyzing the values of density, crystallinity, cellulose, and MFA properties of different NLFs (Table 5), one should notice that TVF is among the lightest. In fact, other authors did not specify whether the density obtained for the fibers compared was absolute or apparent. However, comparing the absolute density of TVF (1.62 g/cm³), similar values are observed. The degree of crystallinity was also within the expected range for the NLFs, as well as the MFA, which are factors directly associated with fiber strength. On the other hand, the theoretically calculated cellulose percentage was lower than that of the other natural fibers presented in Table 4.

**Table 5.** Maximum tensile strength calculated for each diameter interval and Weibull parameters.

| Diameter Range (μm) | $\sigma_{max}$ (MPa) | Standard Deviation | β | θ | $R^2$ |
|---|---|---|---|---|---|
| 356.63–432.03 | 32.665 | 6.078 | 5.459 | 34.610 | 0.940 |
| 432.03–507.43 | 26.175 | 4.708 | 5.426 | 28.270 | 0.959 |
| 507.43–582.83 | 24.206 | 3.302 | 7.299 | 25.700 | 0.913 |
| 582.83–658.23 | 22.675 | 7.981 | 2.493 | 26.110 | 0.847 |
| 658.23–733.63 | 22.369 | 6.832 | 2.993 | 25.280 | 0.923 |
| 733.63–809.03 | 22.226 | 6.832 | 2.983 | 25.300 | 0.704 |
| 809.03–884.43 | 21.670 | 6.761 | 1.785 | 25.420 | 0.962 |
| 884.43–959.83 | 26.341 | 10.394 | 4.082 | 29.010 | 0.898 |
| 959.83–1035.23 | 31.243 | 10.867 | 2.986 | 35.160 | 0.945 |
| 1035.23–1110.63 | 29.750 | 3.167 | 9.211 | 31.240 | 0.853 |

*3.3. Tensile Tests*

Table 5 presents the values of maximum tensile strength obtained in the tensile test of the TVFs, which lists the 10 proposed intervals of diameters.

Based on the verified values of $R^2$, it is observed that the data follow a Weibull statistical distribution with good precision (above 0.85). Furthermore, it is verified that the parameter θ of the Weibull statistic, which gives the characteristic of maximum stress supported by the fiber, was very close to the results obtained in the test. Therefore, high values of θ are indicative of better performance of this material in tensile strength. To understand the results, a graph of maximum tensile strength was plotted by the average diameter of each interval, as shown in Figure 7.

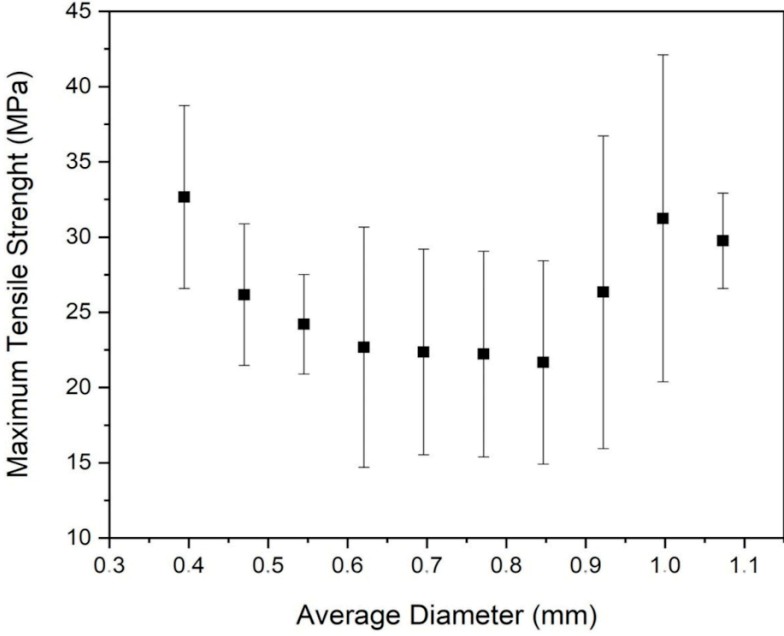

**Figure 7.** Maximum tensile strength of TVFs as a function of diameter.

The average maximum tensile strength for TVFs was calculated at 25.92 MPa. Based on Figure 7, there was an apparently tendency that as the fiber diameter increases, the maximum tension lowers. However, the last three intervals differed from this evaluation, and the ANOVA proved that there are no different averages, as presented in Table 6. In other words, the apparent tendency does not consider the standard deviation. This points out the ANOVA test's importance. The high variability found is one of the main characteristics of a typical natural fiber.

**Table 6.** ANOVA for maximum tensile strength of TVFs for different diameters.

| Variation Causes | DF | SS | MS | $F_{calculated}$ | $F_{critical}$ |
|---|---|---|---|---|---|
| Treatment | 9 | 732.62 | 81.40 | 1.61 | 2.12 |
| Residue | 40 | 2022.32 | 50.56 | | |
| Total | 49 | 2754.94 | | | |

As the $F_{critical}$ is greater than the $F_{calculated}$, it can be stated that the average tensile strength for the tested ranges are statistically equal to a 95% confidence level. Through the tensile test of the fibers, in addition to the property of tensile strength, it was possible to obtain their elastic modulus (Young's modulus) and strain rate. The values of the last two mechanical properties can be evaluated, as shown in Figure 8.

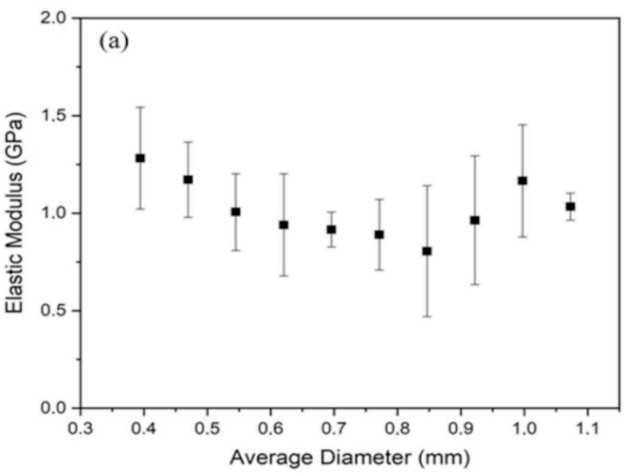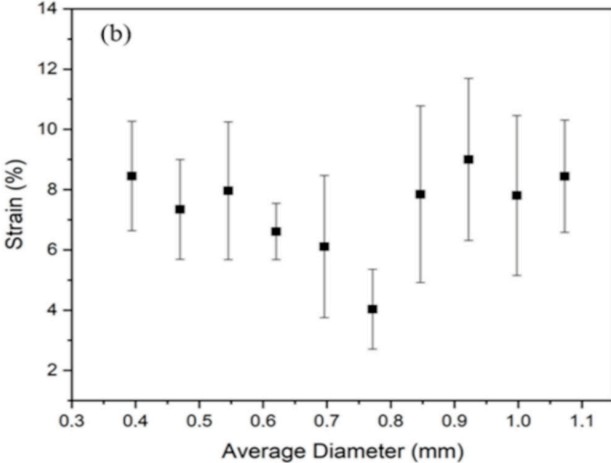

**Figure 8.** (**a**) Elastic modulus; (**b**) strain of TVFs.

Similar to what happened in the tensile strength of the fiber, the Young's modulus (Figure 8), in general, tended to decrease with the increase in its diameter. The exception of this behavior, once again, prevailed in the last three groups. According to Satyanarayana [71], this behavior can be explained by the increase in internal defects being proportional to the growth of the fiber diameter. The strain rate (Figure 8b) presented by the TVFs ranged from 4–9%, showing no specific pattern. Table 7 gathers the average values of the main mechanical properties obtained for the TVFs through tensile test, compared to other previously studied NLFs.

It is known that a higher cellulose content is in association with an increase in tensile strength, as observed in similar NLFs such as kenaf, sugar cane, and others (Table 7). However, through theoretical calculations, it was found that the cellulose content of TVFs was relatively low (39%), which justifies the tensile strength exhibited by this fiber. Furthermore, it is suggested that lignin content might impair the mechanical behavior of the fiber [75]. According to Table 7, it can be note that the TVF presented a relatively low value of tensile strength and elastic modulus, however close to some recently studied NLFs for application as reinforcement of composite materials. Therefore, it is proposed that TVFs can be used in applications requiring a combination of low weight and medium strength.

**Table 7.** Mechanical properties of TVFs compared to other natural fibers.

| Fiber | $\sigma_{max}$ (MPa) | E (GPa) | $\varepsilon$ (%) | Reference |
|---|---|---|---|---|
| Titica vine | $25.92 \pm 6.69$ | $1.02 \pm 0.22$ | $7.36 \pm 2.05$ | PW |
| *Catharanthus roseus* | $27.02 \pm 1.1$ | $1.23 \pm 0.04$ | $2.15 \pm 0.10$ | [64] |
| *Acácia tortilis* | 71.63 | 4.21 | 1.33 | [75] |
| Sugar cane | $169.51 \pm 18.65$ | $5.18 \pm 0.63$ | $6.25 \pm 0.01$ | [76] |
| *Tridax procumbens* | $25.75 \pm 2.45$ | $0.94 \pm 0.09$ | $2.77 \pm 0.64$ | [77] |
| Kenaf | $280 \pm 90$ | $22 \pm 6$ | $1.29 \pm 0.20$ | [78] |
| Coir | $44 \pm 8$ | $2 \pm 0.30$ | $4.5 \pm 0.80$ | [78] |
| *Agave Tequilana Weber Azul* | $68.2 \pm 30$ | $2.39 \pm 0.71$ | $7.40 \pm 4.5$ | [79] |

*3.4. Pullout Test*

Figure 9 presents the pullout test for the different embedded lengths (L) of the TVFs in the epoxy matrix. This graph made it possible to determine the maximum stress in pullout, critical length ($L_c$), and interfacial shear strength ($\tau$).

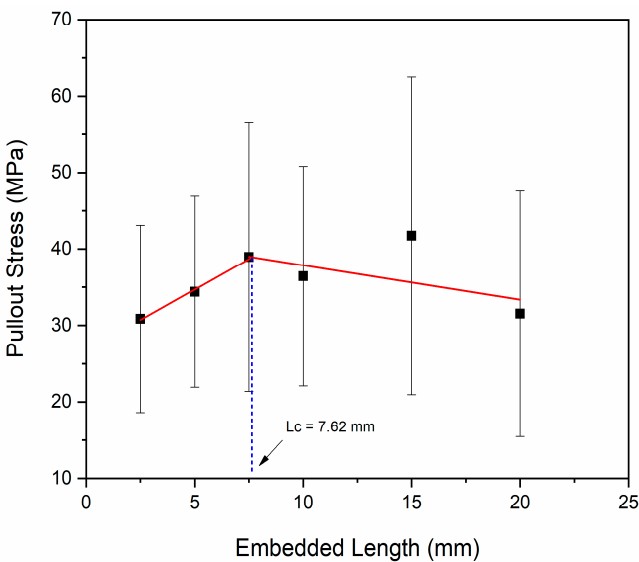

**Figure 9.** Interfacial pullout strength between the epoxy matrix and TVFs, as a function of embedded length.

The result obtained in the test consists of two lines that intersect at the $L_c$ of the TVF, in relation to the epoxy matrix. The first stage is marked by the increase in tensile strength linearly with the length of the fiber embedded in the matrix. This straight line of lesser inclination is represented by the linear adjustment between the values of maximum stress in pullout observed within the range of embeddedness of 2.5 to 7.5 mm, being found around 39 MPa. Equation (9) (highest slope curve) and Equation (10) (lowest slope curve) correspond to the linear adjustments applied to the TVFs pullout stress.

$$\sigma = 1.59L + 26.75 \tag{9}$$

$$\sigma = -0.45L + 42.36 \tag{10}$$

The intersection of Equations (9) and (10) defined the value of $L_c$ = 7.62 mm. From this value, it was possible to calculate the interfacial shear strength, according to Equation (10), which directly influences the mechanical behavior of the composite. For its determination, the value of the mean diameter of the TVFs, 0.6501 mm, which are inserted in interval 4 (range 582.83–658.23 μm), was used. In addition, the average stress of this group, 22.675 MPa, served as the basis for the calculations.

Table 8 presents a comparative study of this value of the interfacial strength of TVF in relation to the other NLFs' reinforcing epoxy matrix.

**Table 8.** Comparison of interfacial shear strength for different NLFs.

| Fiber | Matrix | $\tau$ (MPa) | Reference |
|---|---|---|---|
| TVF | Epoxy | 0.97 | PW |
| Betelnut | Epoxy | 0.88 | [80] |
| Fique | Epoxy | 0.27 | [81] |
| Coir | Epoxy | 1.42 | [82] |
| Palf | Epoxy | 4.93 | [82] |

The value obtained for the interfacial shear strength for the TVF was found to be relatively low (0.97 MPa), but close to that of other NLFs previously studied. According to Luz et al. [82] and Prasad et al. [83], this may be associated with the hydrophilic nature of the fibers. In fact, NLFs have waxy layers and undesirable chemical elements on their surface, which makes the adhesion to the matrix have polar characteristics.

*3.5. Scanning Electron Microscopy (SEM)*

From the SEM analysis, it was possible to reveal the morphologies of the longitudinal and transversal sections of the TVF, as shown in Figure 10. The characteristics evaluated in Figure 10a,b reveal a thin fiber, with a number of defects that are slightly less than the thicker fiber (Figure 10b). These defects are shown in greater detail in Figure 10c,d, as well as the appearance of pores, voids, cracks, microcracks, and fiber bridges.

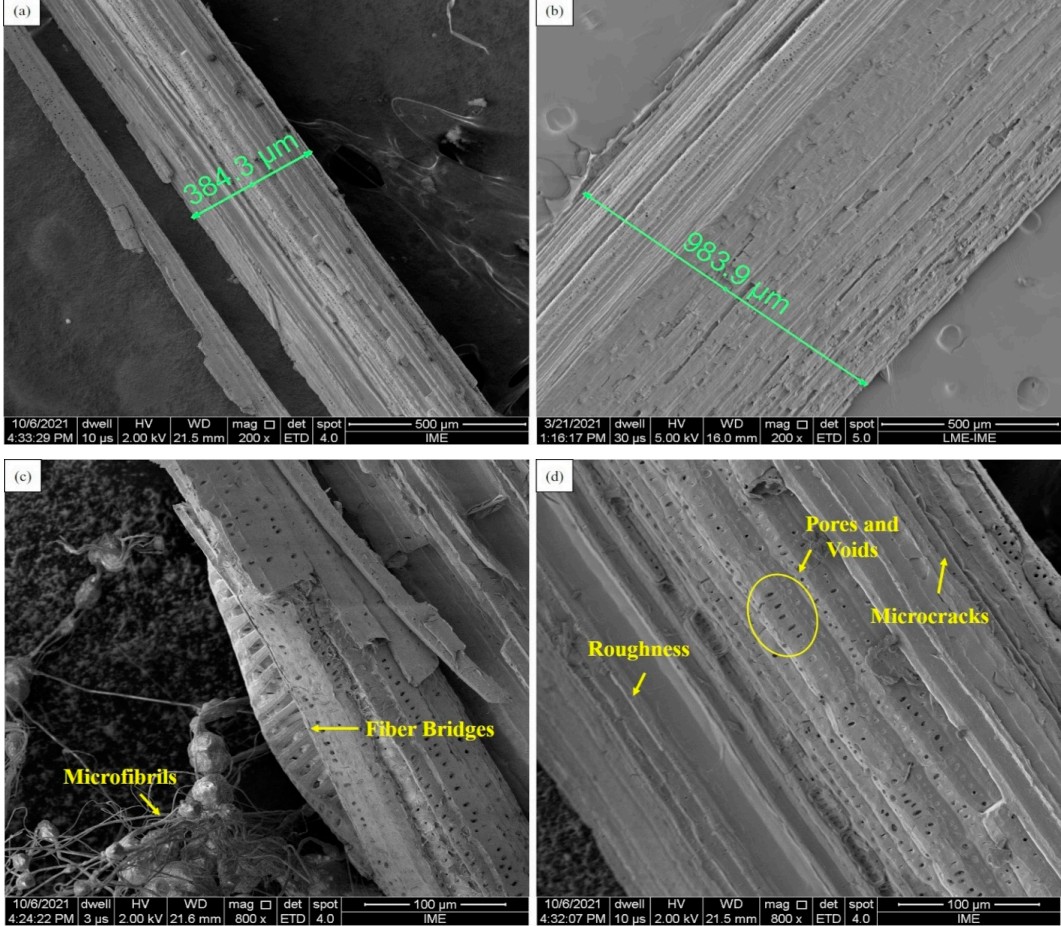

**Figure 10.** *Cont*.

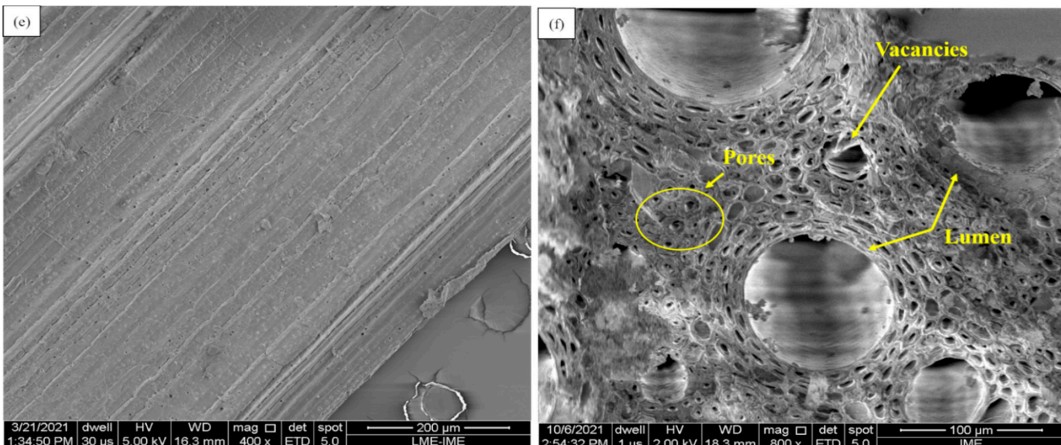

**Figure 10.** SEM images of the morphological surface of the TVF. (**a**) Longitudinal section of a thin fiber; (**b**) longitudinal section of a thick fiber; (**c**,**d**) longitudinal section showing defects in the TVF; (**e**) smooth surface and layer of impurities deposited on the fiber surface; (**f**) cross section showing pores, vacancies, and lumens.

It is possible to observe the variability of the characteristics inherent to these fibers, such as what occurs in Figure 10d, which shows a fiber with a degree of roughness greater than that of Figure 10e, where it has a smooth surface. This fact is probably related to the presence of impurities, waxes, extractives, and residues from the processing and handling processes [84]. Another point that deserves to be mentioned is the presence of the large number of pores, voids, and lumens that are found inside NLFs (Figure 10f). This SEM image of the TVF cross section corroborates the results of the high total and closed porosity calculated previously.

## 4. Summary and Conclusions

A first investigation into the physical, chemical, and mechanical properties of titica vine fiber (TVF) extracted from the root of an Amazonian plant, scientifically called *Heteropsis flexuosa*, revealed favorable and interesting characteristics for the area of polymeric composites with incorporated natural fibers. The study pointed to the following results.

- The TVF density measurements obtained by the geometric method for the 10 diameter intervals ranged from 0.59 to 0.44 g/cm$^3$. The average of the set was around 0.50 g/cm$^3$. The data obtained in the test and by the Weibull statistics revealed a tendency for decreasing density with increasing fiber diameter, from 357 to 1111 μm. Using Archimedes' principle, the fibers showed an average density of 0.57 g/cm$^3$, very close to that found by the geometric method, which is among the lowest for natural fibers. Gas pycnometry returned a density value of 1.62 g/cm$^3$. This high value, in large part, can be justified due to the exclusion of the open porosity of the fiber. However, this is still a similar value when comparing to other NLFs.
- The total, open, and closed porosity for the entire set of evaluated fibers returned values of 69.01%, 11.62%, and 57.39%, respectively. This indicates that TVF is highly porous, being suitable for applications that require lightweight materials.
- The moisture content presented by the TVFs was 11.38%, being considered within the standard for fibers that are commonly used in reinforced polymer composites.
- The FTIR indicated the presence of adsorption bands such as O-H, C-H, and C-O, which are characteristics of NLFs rich in cellulose, lignin, and hemicellulose.
- The fiber's crystallinity index was 78.3%, and the microfibrillar angle (MFA) was approximately 7.95°. Despite the good percentage of crystallinity and the low angle, essential for increasing the fiber's mechanical strength, the theoretical cellulose content calculated was around 39%, which is considered relatively low.

- The maximum tensile strength of the fibers was around 33 to 30 MPa for the 10 intervals of diameters evaluated. The ANOVA showed that there was no influence by the diameter to increase the property. The maximum tensile strength for the whole set was 25.92 MPa, considered relatively low when compared to other traditional natural fibers. This value can be explained by the low percentage of cellulose present in the fiber (39%). The values obtained for elastic modulus and deformation were 1.02 and 7.36, respectively. These results can be compared to other natural fibers under recent study.
- Pullout tests revealed that the critical length of the TVF in the epoxy matrix is 7.62 mm, and the interfacial shear stress 0.97 MPa.
- Through SEM images, it was possible to observe typical microstructural defects in the TVF such as void, microcracks, cracks, and fiber bridges. The high porosity calculated could be observed through the cross-section image of the fibers, which presented many vacancies filled by air and lumens.

Hence, the aerospace, ballistics, and automobile industries, as well as those industries that require projects that consider low density, lower cost, and eco-friendly materials, are potential sectors for the application of composites of TVFs incorporated in a polymer matrix.

**Author Contributions:** Conceptualization, J.d.S.C.d.C. and L.F.C.N.; methodology, J.d.S.C.d.C. and L.F.C.N.; validation, L.F.C.N., F.S.d.L., M.S.O. and S.N.M.; formal analysis, J.d.S.C.d.C.; investigation, J.d.S.C.d.C.; resources, S.N.M. and F.S.d.L.; data curation, J.d.S.C.d.C. and M.S.O.; writing—original draft preparation, J.d.S.C.d.C.; writing—review and editing, F.S.d.L., F.d.C.G.F. and S.N.M.; visualization, L.F.C.N. and F.d.C.G.F.; supervision, L.F.C.N.; project administration, S.N.M.; funding acquisition, S.N.M. All authors have read and agreed to the published version of the manuscript.

**Funding:** This research received no external funding.

**Institutional Review Board Statement:** Not applicable.

**Informed Consent Statement:** Not applicable.

**Data Availability Statement:** Not applicable.

**Acknowledgments:** The authors would like to thank the Brazilian agencies CAPES, CNPq, FAPERJ, and FAPEAM for their support.

**Conflicts of Interest:** The authors declare no conflict of interest.

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
