# Peer review of "Titica Vine Fiber (Heteropsis flexuosa): A Hidden Amazon Fiber with Potential Applications as Reinforcement in Polymer Matrix Composites"

_jcs, doi:10.3390/jcs6090251_

Round 1

Reviewer 1 Report

The paper can be accepted for publication after a revision of English language.

Only few points should be considered to improve the clarity of the paper:

1) It could be useful for the reader of the paper to have a short description of the meaning of the  Weibull "theta" and "beta" parameters, which have been explicitly reported in several tables;

2) line 291: The MFA of the TVF was calculated as 7.95°...

It could be useful for the reader, if the authors breafly clarify how this angle value is obtained by the graphic analysis of Fig. 6.

3) lines 310-311: the sentence "In addition, it appears that the property evaluated by the Weibull statistics was very close to the one obtained in the test" should be clarified.

Author Response

Response to Reviewers – TVF/JCS 2022

The authors would like to thank the reviewers for the valuable comments and suggestions that contribute to improve the manuscript. Responses to each comment are listed below and all modifications/additions were marked as Track Changes in the revised version of the manuscript.

Reviewer  #1

General comment: The paper can be accepted for publication after a revision of English language.

Response: The authors are thankful to the reviewer indication. As requested, the English language improvement was made in the new revised version of the manuscript.

Comment (1): It could be useful for the reader of the paper to have a short description of the meaning of the Weibull "theta" and "beta" parameters, which have been explicitly reported in several tables.

Response: Indeed, the reviewer is right. The descriptions of “theta” and “beta” parameters were inserted in the revised version, specifically in section 3.1.1.

Comment (2): line 291: The MFA of the TVF was calculated as 7.95°... It could be useful for the reader, if the authors breafly clarify how this angle value is obtained by the graphic analysis of Fig. 6.

Response: As requested by the reviewer, the correlation between fiber MFA and its influence on mechanical properties was introduced. in the revised version.

Comment (3): lines 310-311: the sentence "In addition, it appears that the property evaluated by the Weibull statistics was very close to the one obtained in the test" should be clarified.

Response: The observation made by the reviewer is correct, so that the relationship between the parameters of the Weibull statistics and the data obtained in the test was clarified.

Reviewer 2 Report

The manuscript introduced a little-known natural fiber, Titica Vine Fiber (TVF), as a composite filler material. The authors focused on figuring out the density of the TVF then they postulated the obtained value (0.50 g/cm3) is the lowest density compared to other natural fibers to date. The authors also carried out typical measurements to figure out porosity, composition, crystallinity, and tensile properties. Although the authors performed various measurements to clarify basic information of the rarely used natural fiber, I am not sure about the novelty of the work. The authors already recognized the pioneering work that prepared TVF/epoxy matrix and characterized the performance of the composite, and claimed the difference between this manuscript and the previous one [Polymers, 2021, 13, 4079]; however, I believe their claim is somewhat limited. This manuscript seems to be supplemental information of the previous paper. For the publication in this journal, the authors should display at least a composite behavior with this natural fiber information as a “reinforcement in polymer matrix composites” in title. They tried to prove the composite homogeneity and miscibility between the polymer matrix and the fiber filler from hydrophobicity of the natural fiber, there are however many parameters to determine the composite properties. Hence I recommend the authors have to show some properties of the polymer matrix/TVF, at least they have to prove adhesion between the materials. I also prefer to revise something in this manuscript to make the readers understanding clear.

1)      The authors provided the lowest density of the TVF from the first and second method, although the third method showed normal density with convincing reason (gas pycnometer was limited to detect closed pore). Could you show which method was used to figure out the densities in previous studies in Table 4? If all the previous studies used the third method, the novelty of this work might be attenuated.

2)      The authors provided an example that showed low density (Curaua). How about its properties? Could you provide some mechanical and other information of the low-density natural fibers?

3)      There is no information to explain abbreviation, especially, related to Weibull parameters.

4)      I agree the Gaussian fitting to figure out crystalline of the TVF. But the explanation is limited how to calculate the angle and what experimental setup was used to measure XRD. Did the authors put fibers on the x-ray spot with well-aligned state or make powder?

Author Response

Response to Reviewers – TVF/JCS 2022

The authors would like to thank the reviewers for the valuable comments and suggestions that contribute to improve the manuscript. Responses to each comment are listed below and all modifications/additions were marked as Track Changes in the revised version of the manuscript.

Reviewer  #2

General comment: The manuscript introduced a little-known natural fiber, Titica Vine Fiber (TVF), as a composite filler material. The authors focused on figuring out the density of the TVF then they postulated the obtained value (0.50 g/cm3) is the lowest density compared to other natural fibers to date. The authors also carried out typical measurements to figure out porosity, composition, crystallinity, and tensile properties. Although the authors performed various measurements to clarify basic information of the rarely used natural fiber, I am not sure about the novelty of the work. The authors already recognized the pioneering work that prepared TVF/epoxy matrix and characterized the performance of the composite, and claimed the difference between this manuscript and the previous one [Polymers, 2021, 13, 4079]; however, I believe their claim is somewhat limited. This manuscript seems to be supplemental information of the previous paper. For the publication in this journal, the authors should display at least a composite behavior with this natural fiber information as a “reinforcement in polymer matrix composites” in title. They tried to prove the composite homogeneity and miscibility between the polymer matrix and the fiber filler from hydrophobicity of the natural fiber, there are however many parameters to determine the composite properties. Hence I recommend the authors have to show some properties of the polymer matrix/TVF, at least they have to prove adhesion between the materials. I also prefer to revise something in this manuscript to make the readers understanding clear.

Response: The authors are thankful for the reviewer’s appreciation words regarding our manuscript. The mentioned points were taken into consideration. The pullout test was added to the revised manuscript, as can be found now in section 3.4. The method involves pulling a single fiber partially embedded in an epoxy matrix. This test was able to verify the adhesion of the TVF in epoxy matrix, as well as to determine the critical fiber length and the interfacial shear strength. In this way, in the hope of to provide the demand required by the reviewer regarding the request of relevant parameters considering the composite (TVF/Epoxy).

Comment (1): The authors provided the lowest density of the TVF from the first and second method, although the third method showed normal density with convincing reason (gas pycnometer was limited to detect closed pore). Could you show which method was used to figure out the densities in previous studies in Table 4? If all the previous studies used the third method, the novelty of this work might be attenuated.

Response: Previous studies used methods to calculate the apparent density (geometric method or Archimedes). This information was more clearly evidenced in the new revised manuscript.

Comment (2): The authors provided an example that showed low density (Curaua). How about its properties? Could you provide some mechanical and other information of the low-density natural fibers?

Response: Mechanical properties of curaua fiber and other low-density fibers were included in the revised new manuscript.

Comment (3): There is no information to explain abbreviation, especially, related to Weibull parameters.

Response: In fact, the reviewer is right, so the description of abbreviations used was added, as well as the Weibull parameters.

Comment (4):  I agree the Gaussian fitting to figure out crystalline of the TVF. But the explanation is limited how to calculate the angle and what experimental setup was used to measure XRD. Did the authors put fibers on the x-ray spot with well-aligned state or make powder?

Response: Excellent observation made by the reviewer. In this way, the analysis details information has been added. As for XRD analysis, vertically aligned fibers were used. A better description of how to calculate the MFA was inserted in the revised manuscript as well.

Round 2

Reviewer 1 Report

The authors have integrated the revised version of the manuscript as required.

Their paper can be published in the present form, after some further minor language corrections that can be easily inserted in the page proofs during the last steps of the editorial process . 

Reviewer 2 Report

This article is sufficiently changed, then it may be published at this journal.